# Upregulation of the Long Noncoding RNA CASC10 Promotes Cisplatin Resistance in High-Grade Serous Ovarian Cancer

**DOI:** 10.3390/ijms23147737

**Published:** 2022-07-13

**Authors:** Ricardo Noriega-Rivera, Mariela Rivera-Serrano, Robert J. Rabelo-Fernandez, Josué Pérez-Santiago, Fatima Valiyeva, Pablo E. Vivas-Mejía

**Affiliations:** 1Department of Biochemistry, Medical Sciences Campus, University of Puerto Rico, San Juan, PR 00936, USA; ricardo.noriega1@upr.edu; 2Comprehensive Cancer Center, Medical Sciences Campus, University of Puerto Rico, San Juan, PR 00936, USA; mariela.rivera20@upr.edu (M.R.-S.); robert.rabelo@upr.edu (R.J.R.-F.); joperez@cccupr.org (J.P.-S.); fvaliyeva@cccupr.org (F.V.); 3Department of Biology, Rio Piedras Campus, University of Puerto Rico, San Juan, PR 00931, USA; 4School of Dental Medicine, Medical Sciences Campus, University of Puerto Rico, San Juan, PR 00936, USA

**Keywords:** ovarian cancer, cisplatin resistance, RNA-seq, bioinformatics, long noncoding RNAs

## Abstract

Despite initial responses to first-line treatment with platinum and taxane-based combination chemotherapy, most high-grade serous ovarian carcinoma (HGSOC) patients will relapse and eventually develop a cisplatin-resistant fatal disease. Due to the lethality of this disease, there is an urgent need to develop improved targeted therapies against HGSOC. Herein, we identified CASC10, a long noncoding RNA upregulated in cisplatin-resistant ovarian cancer cells and ovarian cancer patients. We performed RNA sequencing (RNA-seq) in total RNA isolated from the HGSOC cell lines OVCAR3 and OV-90 and their cisplatin-resistant counterparts. Thousands of RNA transcripts were differentially abundant in cisplatin-sensitive vs. cisplatin-resistant HGSOC cells. Further data filtering unveiled CASC10 as one of the top RNA transcripts significantly increased in cisplatin-resistant compared with cisplatin-sensitive cells. Thus, we focused our studies on CASC10, a gene not previously studied in ovarian cancer. SiRNA-mediated CASC10 knockdown significantly reduced cell proliferation and invasion; and sensitized cells to cisplatin treatment. SiRNA-mediated CASC10 knockdown also induced apoptosis, cell cycle arrest, and altered the expression of several CASC10 downstream effectors. Multiple injections of liposomal CASC10-siRNA reduced tumor growth and metastasis in an ovarian cancer mouse model. Our results demonstrated that CASC10 levels mediate the susceptibility of HGSOC cells to cisplatin treatment. Thus, combining siRNA-mediated CASC10 knockdown with cisplatin may represent a plausible therapeutic strategy against HGSOC.

## 1. Introduction

Ovarian cancer remains a significant cause of cancer-related deaths, with approximately 19,880 new cases and 12,810 deaths predicted for 2022 in the US alone [1]. Epithelial ovarian carcinoma (EOC) is the most common ovarian cancer type representing 90% of these malignancies [1]. High-grade serous ovarian cancer (HGSOC) represents 70% of all EOCs [2]. The standard line of treatment for ovarian cancer usually consists of cytoreductive surgery combined with chemotherapy with platinum (i.e., cisplatin) and/or taxane-based compounds [3]. Cisplatin forms adducts with the DNA, which causes the inhibition of replication and transcription, and leads to cell-cycle arrest and apoptosis [4]. Cisplatin is chemically inert; however, it becomes activated once inside the cell. Water molecules displace chloride atoms on the cisplatin molecule, and the hydrolyzed product acts as a potent electrophile that can react with any nucleophile (i.e., sulfhydryl groups on proteins and nitrogen donor atoms on nucleic acids) [5]. Specifically, cisplatin binds to the N7 reactive center on purine residues and triggers deoxyribonucleic acid (DNA) damage, resulting in platinum-DNA adducts that compromise the integrity of the DNA and lead to cell division impairment and activation of programmed cell death [6]. In addition, cisplatin alters the mitochondrial membrane potential by inducing oxidative stress and preventing calcium uptake due to a significant loss in protein-SH groups [7].

Initial response rates are 60–80%, but around 70% of HGSOC patients develop a cisplatin-resistant-fatal disease [8]. Postulated mechanisms of cisplatin resistance include decreased levels of receptors/channels that reduce the influx of cisplatin inside cells, increased levels of proteins/channels that promote cisplatin efflux, increased intracellular levels of certain sulfur-containing macromolecules that reduce the nuclear net cisplatin concentration, overexpression of cellular scavenger and detoxification enzymes such as GSH and MTs (glutathione and metallothionein), increased DNA repair process by overexpression of the nucleotide excision repair pathway-related proteins ERCC1 and XPF, and the metabolic rewiring which confers growth advantages to particular cell populations [9,10,11,12]. Studies indicate that inactivation of intrinsic cell death pathways, activation of cell survival pathways, and dysregulation of oncogenes, tumor suppressor genes, and noncoding RNAs may also play a central role in cisplatin resistance of cancer cells [13,14,15]. Nevertheless, the major contributing factors to cisplatin resistance in ovarian cancer cells have not been fully identified.

To further understand the underlying mechanisms of cisplatin resistance in HGSOC and therefore identify targets for therapeutic strategies, we characterized the transcriptome of the HGSOC cell lines OV-90, OVCAR3, and their cisplatin-resistant counterparts. Differentially abundant RNA transcripts were filtered using bioinformatics tools, the Kaplan-Meier plotter cancer patient database, real-time PCR, and a siRNA screening. A total of 27 upregulated RNA transcripts were identified in cisplatin-resistant as compared with cisplatin-sensitive cells. From this list, the non-previously studied long-noncoding RNA (lncRNA) CASC10 was selected for further investigation. We studied the biological and functional effects of siRNA-mediated CAS10 knockdown in cell survival, apoptosis, cell cycle progression, and tumor growth using an ovarian cancer mouse model. Lastly, we identified many downstream effectors following siRNA-mediated CASC10 knockdown in HGSOC cells. Together, these data identified CASC10 as a potential therapeutic target for cisplatin-resistant ovarian cancer treatment.

## 2. Results

### 2.1. Identification of Differentially Expressed Genes in Cisplatin-Sensitive versus Cisplatin-Resistant HGSOC Cells

The concentration of cisplatin inhibiting 50% of cell growth (IC_50_) of the ovarian cancer cell lines has been reported and summarized in Appendix A [16]. To identify critical genes associated with cisplatin resistance, we performed RNA-seq in OV-90 and OVCAR3 and their cisplatin-resistant counterparts OV-90CIS and OVCAR3CIS cells. A diagram of how the RNA-seq data was processed is shown in Figure 1A. We identified 10,714 differentially abundant RNA transcripts (DATs) in OV-90/OV-90CIS and 5328 in OVCAR-3/OVCAR3CIS (Figure 1B). At the first stage of data filtering, we retained genes that were identified as DATs in both pairs of cell lines (n = 5700). DATs identified in only one pair of cell lines were removed. The second filter was introduced to remove those transcripts that showed opposite expression patterns in both cell lines (i.e., upregulated in one pair of cell lines and downregulated in the other pair of cells), which reduced the list to 3749 dysregulated transcripts. Next, we performed a third filter based on the distributions of the base mean intensity of the transcript and fold change. We ranked each transcript (Rank range: 2–8) by adding the numbers corresponding to the quartile of the distribution where the value of both the base mean intensity and fold change lie within the distribution (1–4 for each distribution). For this analysis, we selected all transcripts with ranks ≥ 7, which reduced the number of transcripts from 3749 to 414 (237 upregulated and 177 downregulated in both cell pairs); see Appendix A for the list of the 414 transcripts. Volcano plots showed several transcripts with significant fold changes among the cisplatin-resistant and cisplatin-sensitive cells (Figure 1C). To visualize molecular interactions between the deregulated transcripts, the list with the 414 transcripts was subjected to Ingenuity Pathway Analysis (IPA), resulting in 25 different networks (Appendix A). The top network in the list includes genes involved in anti-apoptotic pathways such as the growth arrest-specific 1 (*GAS1*), vascular endothelial growth factor C (*VEGFC*), potassium sodium-activated channel subfamily T member 2 (*KCNT2*), and mitogen-activated protein kinase 1 (*MAPK1*) (Figure 1D). The second network included synaptotagmin like 2 (*SYTL2*), mitogen-activated protein kinase 1 (*MAPK1* or *ERK*), ATP binding cassette subfamily A member 3 (*ABCA3*), and protein kinase cGMP-dependent 1 (*PRKG1*), which are associated with molecule and vesicle trafficking, downstream phosphorylation, and cGMP signaling (Figure 1E).

To select potential clinically relevant genes associated with the progression of ovarian cancer, we interrogated the KM plotter searchable patient database. The KM plotter includes data from “The Cancer Genome Atlas” (TCGA) data portal and other patient databases for a total of 1656 ovarian cancer samples. We assessed the correlation between the 414 genes and the Overall Survival (OS) and Progression-Free Survival (PFS). Using the KM plotter database, 61 genes showed a significant (*p* < 0.05) difference in the OS and/or PFS. Figure 2A shows the Kaplan-Meier curves for the top four relevant genes (*CASC10*, *PDLIM3*, *EMP1*, and *ATP11B*) of the list. A strong correlation between the RNA expression levels and the OS and the PFS was observed for the four genes shown in Figure 2(Aa–Ad). The Kaplan-Meier curves for the other 57 genes are included in Appendix A. The differential expression levels of the 61 genes were validated by real-time PCR. As shown in Table 1, 45 out of the 61 genes were validated by PCR, of which 28 were upregulated, and 17 were downregulated in OVCAR3CIS when compared to its sensitive counterpart (Figure 2B).

We then performed an RNAi screening by transiently transfecting the OVCAR3CIS cells with a pool of four specific siRNAs against each of the 27 genes, followed by colony formation assays (Figure 2C). We observed more than 50% reductions in the number of colonies for *CASC10*, *ATP11B*, *EMP1*, *GAS1*, *SLC6A15*, *GALNT13*, and *PDLIM3* compared with cells transfected with a NC-siRNA (Figure 2C).

### 2.2. CASC10 Is Upregulated in Ovarian Cancer Patients and Cisplatin-Resistant Ovarian Cancer Cells

According to survival analysis, CASC10 showed the strongest significant correlation between the OS (*p* = 1.6 × 10^−9^ HR = 1.78) and PFS (*p* = 1.5 × 10^−5^ HR = 1.57) of the disease (Figure 2(Aa)). A comparative expression (tumor vs. normal tissue) plot using the Gene Expression Profiling Interactive Analysis (GEPIA) searchable database (RNA-seq data) revealed a statistically significant higher CASC10 expression in ovarian tumors compared to normal ovaries (Figure 2D).

We confirmed the expression of CASC10 by real-time PCR in a panel of ovarian cancer cell lines. The CASC10 levels were higher in the cisplatin-resistant as compared with their cisplatin-sensitive counterparts (Figure 2E). There were no significant differences between A2870 and A2780CP20 cells. Of note, the A2780 cell line is derived from an endometrioid ovarian adenocarcinoma [17].

### 2.3. CASC10 siRNA-Mediated Knockdown Reduced Cell Growth, Invasion, and Viability in Ovarian Cancer Cells

We next studied the biological consequences of siRNA-mediated CASC10 silencing in ovarian cancer cells. The 2^−ΔΔCt^ analysis of an RT-qPCR experiment showed that transient transfection of OVCAR3CIS cells with CASC10-targeted siRNAs decreased the CASC10 expression by 47% with the CASC10-siRNA(1) and in 57% with CASC10-siRNA(2) as compared with the NC-siRNA (**** *p* < 0.001, Figure 3A). In a colony formation assay with OVCAR3CIS, both CASC10-targeted siRNAs reduced the number of colonies formed compared with NC-siRNA transfected cells (Figure 3B). Notably, the CASC10-siRNA(2) reduced the number of OVCAR3CIS colonies by 54% (*** *p* < 0.0001), whereas CASC10-siRNA(1) reduced the number of colonies by only 42% (** *p* < 0.001). We assessed the effect of CASC10 knockdown on the invasion ability of OVCAR3CIS cells. Invasion assays showed that CASC10-siRNA(1) and CASC10-siRNA(2) significantly reduced the invasiveness of OVCAR3CIS (42% reduction; **** *p* < 0.0001 and 62% reduction; **** *p* < 0.0001, respectively) compared with NC-siRNA transfected cells (Figure 3C,D).

In order to assess the effects of CASC10 knockdown in a different type of OC cells other than HGSOC, we performed the CASC10 knockdown SKOV3ip1, which is classified as a clear cell ovarian carcinoma cell line [18]. The 2^−ΔΔCt^ analysis showed a decrease of CASC10 relative expression of 70% (**** *p* < 0.0001) and 75% (**** *p* < 0.0001) following transfection of SKOV3ip1 cells with CASC10-siRNA(1) and CASC10-siRNA(2) respectively (Appendix A) [19]. Interestingly, CASC10-siRNA(2) reduced the number of colonies by 86% (**** *p* < 0.0001), whereas CASC10-siRNA(1) reduced the number of colonies by only 81% (**** *p* < 0.0001) (Appendix A). CASC10 knockdown in SKOV3ip1CIS reduced the invasion ability of these cells in 36% (** *p* < 0.01), and 58% (*** *p* < 0.0001) with the CASC10-siRNA(1) and CASC10-siRNA(2), respectively (Appendix A).

We also investigated whether CASC10-targeted siRNAs alone or in combination with CIS reduced cell viability. The NC-siRNA did not reduce the cell viability of OVCAR3CIS cells at any of the assessed concentrations (Figure 2E). CIS (2.5 μM final concentration) neither reduce the cell viability of NC-siRNA-transfected cells. Transient transfections of 50 nM and 100 nM (final concentrations) of CASC10-siRNA(2) into OVCAR3CIS significantly reduced (15% with 50 nM ** *p* < 0.001 and 30% with 100 nM, ** *p* < 0.001) cell viability compared with the NC-siRNA (Figure 3E). Outstandingly, the combination of CASC10-siRNA(2) with CIS (2.5 μM) significantly reduced to 56% (** *p* < 0.0001) the cell viability compared with NC-siRNA (Figure 3E). Similar cell viability results were obtained when combining CASC10-targeted siRNA with CIS in SKOV3ip1CIS cells (Appendix A). We also performed cell viability and cell invasion experiments transfecting CASC10-targeted siRNA in OVCAR3 and SKOV3ip1 cells. The CASC10-targeted siRNA (2) did not significantly reduce cell viability or invasion at any siRNA concentrations tested compared with the NC-siRNA (Appendix A).

### 2.4. CASC10 siRNA-Mediated Knockdown Induced Apoptosis and Cell Cycle Arrest

We further studied whether the reduction in cell growth and proliferation after CASC10 knockdown was due to activation of apoptosis, cell cycle arrest, or both. Compared to NC-siRNA, siRNA-mediated CASC10 knockdown in OVCAR3CIS cells resulted in a 5-fold increase in caspase-3 activity (** *p* = 0.0016, Figure 4A). Similar results were obtained for SKOV3ip1CIS (4-fold increase; ** *p* < 0.0016, Appendix A). Activation of apoptosis was confirmed when we assessed the changes in apoptotic-related proteins by western blot analysis. Cells treated with CASC10-siRNA(2) showed a significant increase in the active form of Caspase-9 (cleaved Caspase-9) and Caspase-3 (cleaved Caspase-3) (* *p* < 0.05 and * *p* < 0.05, respectively, Figure 4B,C). A significant increase in the cleaved poly-ADP ribose polymerase-1 (PARP-1) was also observed in CASC10-siRNA(2) as compared with NC-siRNA transfected cells (**** *p* < 0.0001, Figure 4C). Moreover, we observed a strong reduction of the anti-apoptotic protein, Bcl-2, following siRNA-mediated siRNA knockdown compared with NC-siRNA-transfected cells (** *p* = 0.022, Figure 4D). Similar results were observed for SKOV3ip1CIS (2-fold cleaved Caspase-9 increase; *** *p* = 0.0005, 6-fold cleaved Caspase-3 increase; ** *p* = 0.0076, and 7-fold cleaved PARP-1 increase; **** *p* < 0.0001, 66% Bcl-2 decrease; *** *p* = 0.0003) (Appendix A).

The effect of siRNA-mediated CASC10 knockdown on cell cycle progression was assessed by flow cytometry. A dramatic cell cycle arrest in the G0/G1 to S phase was observed in OVCAR3CIS and SKOV3ip1CIS 48 h post-transfection (**** *p* < 0.0001, and **** *p* < 0.0001, respectively, Figure 4E,F and Appendix A). These results were confirmed by western blot, where we observed changes in key proteins involved in the G0/G1 to S phase checkpoint. Intriguingly, we observed a reduction in the protein levels of the tumor suppressor p27 in OVCAR3CIS-CASC10-siRNA(2) and SKOV3ip1CIS-CASC10-siRNA(2) compared with NC-siRNA-transfected cells (**** *p* < 0.0001, and *** *p* = 0.0002 respectively, Figure 4G–I and Appendix A). In addition, a reduction of the checkpoint proteins of the S phase, Cyclin E1 and CDK4 was observed in OVCAR3CIS-CASC10-siRNA(2) and SKOV3ip1CIS-CASC10-siRNA(2) compared with NC-siRNA-transfected cells (** *p* = 0.0017, * *p* < 0.03 and *** *p* = 0.0001, *** *p* = 0.0007 respectively, Figure 4G–I and Appendix A).

### 2.5. In Vivo Targeting of CASC10 with Liposome-Encapsulated siRNAs

Next, we asked whether the siRNA-mediated CASC10 knockdown reduced in vivo tumor growth. We encapsulated siRNAs into DOPC-based nanoliposomes. We have extensively characterized and used siRNA-liposomal formulations for siRNA delivery in mouse models [20]. The tumor weight and the number of nodules were significantly lower in the CASC10-siRNA group in comparison with the NC-siRNA or with the cisplatin groups (* *p* < 0.05, Figure 5A,B). These effects were exacerbated when CASC10-siRNA was combined with cisplatin (** *p* < 0.008 Figure 5A,B). A visual image of tumor size in all conditions is shown in Figure 5C. We did not observe any weight differences among the different groups of mice at the end of the experiment (Figure 5D). In summary, combination therapy of liposomal CASC10-siRNA and CIS attenuated tumor progression in a cisplatin-resistant mouse model of HGSOC.

### 2.6. Downstream Effectors of CASC10 in HGSOC Cells

CASC10 is a long noncoding RNA (antisense lincRNA) with a length of 3799 bp located in the reverse strand of chromosome 10 (Figure 6A) [21]. Splicing of the transcribed RNA produces a 3799 bp by elimination of an intronic region of 804 bp. Neither the biological role nor the cellular localization of this noncoding RNA is currently known. We used the LncATLAS, a web-based cell visualization tool (https://lncatlas.crg.eu/, accessed on 15 February 2022) that uses available subcRNA-seq raw data from 15 well-known cell lines from the ENCODE consortium and quantifies the RNA localization using the “relative concentration index” (RCI). RCI is defined as the log_2_-transformed ratio of FPKM (fragments per kilobase per million mapped) in two samples (i.e., nucleus and cytoplasm) [22]. This analysis showed that the expression of CASC10 is higher in the nuclear fraction than the cytoplasmic fraction in 10 of the 15 well-known cell lines (Figure 6B). In addition, we observed a significant enrichment of CASC10 RNA levels in the chromatin subcompartment in the nucleus of K562 cells (Appendix A).

To gain further insights into the signaling pathways downstream of CASC10, we carried out a transcriptome-wise analysis by RNA sequencing (RNA-seq) after siRNA-mediated CASC10 knockdown in OVCAR3CIS cells. Using an initial *p*-adjusted value (*padj*) cutoff <0.01, we identified 1560 differentially abundant RNA transcripts (DATs) between NC-siRNA and CASC10-siRNA(2). One hundred sixty transcripts were regulated in NC-siRNA as compared with non-treated cells, which could represent off-target siRNA effects (see the Venn diagram, Figure 6C). In total, 1400 differentially abundant transcripts were exclusive of CASC10-siRNA(2) compared with NC-siRNA, including 736 downregulated and 816 upregulated transcripts.

The 1400 differentially abundant RNA transcripts (DATs) were used to analyze functional enrichment using Metascape via Gene Ontology (GO) and the Kyoto Encyclopedia of Genes and Genomes (KEGG). The top 20 most significantly (*p*-value ≤ 0.01) enriched ontology clusters include mitotic cell cycle processes, histone modifications, cell cycle, mRNA metabolic processes, cellular response to stress, and cellular response to DNA damage stimulus (Figure 6D). In addition, we used Metascape to identify transcriptional regulatory transcription factors (TFs) for the identified DEGs [23]. Most enriched ontology clusters were regulated by transcription factors such as *E2F1*, *EGR1*, *E2F3*, *TP53*, *SOX6*, *NFYA*, and *SIRT1* (Figure 6E).

A further log2 fold change cutoff >1.2 or <−1.2 with a *p*-value ≤ 0.01 was used to select the most relevant differentially expressed genes following CACS10 knockdown. Applying these criteria, we identified 32 differentially expressed genes, 18 upregulated and 14 downregulated in CASC10-siRNA(2) vs. NC-siRNA transfected cells (Table 2). Based on these criteria, among the upregulated genes, the top five included *RTN4R*, *KIAA0754*, *PYM1*, *CNN1*, and *TGFBRAP1*. The top five of the 14 downregulated genes include *NUP43*, *FHL1*, *DHFR2*, *MIR1915HG*, and *NDUFA7* (Table 2). To better visualize the molecular interactions between the 32 differentially abundant transcripts, we performed IPA. The top network in the list includes genes involved in cell death and survival pathways such as *Cyclin D*, *MERTK*, *TNF*, and *CDK4* (Figure 6F). In addition, the top canonical pathways involved HER-2 signaling in breast cancer, cell cycle Regulation by BTG family proteins, cell cycle control of chromosomal replication, and PTEN signaling (Appendix A).

## 3. Discussion

High-grade serous ovarian cancer (HGSOC) is highly associated with disease recurrence and platinum resistance. Although multiple molecular pathways contributing to cisplatin resistance have been identified, not optimal therapies against cisplatin-resistant ovarian cancer are currently available [24]. In the present study, we performed a systematic and comprehensive identification of dysregulated RNAs in cisplatin-resistant as compared with cisplatin-sensitive HGSOC cells. RNA-seq followed bioinformatics, OS, and PFS KM curves, and an RNAi screening identified several potential genes for ovarian cancer therapy. Particularly, siRNA-mediated knockdown of seven genes, *CASC10, ATP11B, EMP1, GAS1, SLC6A15, GALNT13,* and *PDLIM3*, significantly reduced cell proliferation of ovarian cancer cells. Moreno-Smith et al. showed that ATP11B is overexpressed in human ovarian cancer samples and cisplatin-resistant ovarian cancer cell lines [25]. Elevated ATP11B levels promoted the export of cisplatin from cells [25]. Liu et al. showed that EMP1 was upregulated in ovarian cancer cell lines and tissues and facilitated cell proliferation, invasion and EMT through the RAS/MAPK/c-JUN pathway [26]. Bignotti et al. showed evidence that PDLIM3 is overexpressed in ovarian serous papillary carcinoma (OSPC) and metastatic ovarian serous papillary carcinoma (MET) and is a potential therapeutic marker [27].

CASC10, also known as MIR1915HG, is a lncRNA of unknown cellular localization and function. LncRNAs molecules play important roles at every step of the gene expression course, including regulation of transcription, posttranscriptional processing, genomic imprinting, chromatin modification, and regulation of protein function [28,29,30,31,32]. Dysregulation of lncRNAs has been associated with cancer initiation, progression, tumor maintenance, and drug resistance in virtually every tumor type [33]. For example, Wang et al. found 11 lncRNAs (*HOTAIR*, *TC010441*, *ABO73614*, *ANRIL*, *MALAT1*, *NEAT1*, *CCAT2*, *UCA1*, *HOXA11-AS*, *SPRY4-IT1*, and *ZFAS1*) with a significant overexpression in ovarian cancer patients [34]. Here we reported for the first time that CASC10 is increased in ovarian cancer samples compared with control ovaries and in cisplatin-resistant ovarian cancer cells compared with cisplatin-sensitive cells counterparts. CASC10 belongs to the CASC family of lncRNAs which are spread throughout the genome. Shi et al. demonstrated that CASC15, a long noncoding RNA, was significantly downregulated in ovarian cancer tissues and cells [35]. The low expression of CASC15 was closely associated with shorter overall survival (OS) and progression-free survival (PFS) of ovarian cancer patients [36]. Evidence indicates that other members of the CASC family (CASC2, CASC11, CASC9) could be associated with enhanced or reduced proliferation, invasion, and apoptosis in cervical and colorectal cancers [37,38,39].

We observed that siRNA-mediated CASC10 knockdown strongly reduced cell proliferation, viability, and the invasiveness potential of cisplatin-resistant ovarian cancer cells. Most important, CASC10 knockdown fully sensitized ovarian cancer cells to cisplatin treatment. Furthermore, we observed strong increases in caspase-3 activity and other apoptotic markers, which suggest that CASC10 controls cell survival-related molecules. Interestingly, we observed cell cycle arrest specifically in the G0/G1 to S phase transition following CASC10-siRNA knockdown. These results were then confirmed by the observed reduction in the Cyclin E1 and CDK4 protein levels upon CASC10 knockdown. Intriguingly, we observed reduced protein levels of the cell cycle inhibitory protein p27 following CASC10 knockdown. Increased levels of p27 are normally observed upon cell cycle arrest under several conditions [40]. However, our studies are in agreement with studies of Hiromura et al. on mesangial cells and fibroblast [41]. They observed increased apoptosis when the levels of p27 were absent or reduced [41]. This information suggests that p27 is able to trigger proliferation or cell death depending on the presence or absence of mitogenic signals. In addition, cellular stress conditions that are induced by chemotherapeutic agents, growth factor deprivation, and reduced or absence of p27 levels increase apoptosis, an effect mediated by CDK2 [42].

To date, the standard line of treatment for ovarian cancer includes the use of cisplatin and paclitaxel, but unfortunately, many patients develop chemoresistance leading to therapeutic failure [43]. Our liposomal CASC-10-siRNA formulation reduced tumor growth and metastasis in an HGSOC mouse model. This effect was exacerbated when the liposomal formulation was combined with cisplatin. Thus, our study provides further evidence that CASC10 is a plausible target for ovarian cancer treatment. Additional in vivo studies combining different doses of CASC10-siRNA and cisplatin will confirm the synergistic interaction of both drugs in suppressing tumor growth. Further pharmacokinetics, pharmacodynamics, and safety studies are also needed before moving this formulation into clinical trials. Importantly, siRNA-mediated CASC10 knockout in cisplatin-sensitive cells did not induce noticeable changes in cell proliferation, suggesting that targeting this molecule will not produce off-target or undesirable effects in other cell types. This hypothesis should be confirmed with additional pre-clinical studies.

Interrogation of databases suggests that CASC10 is located in the nuclear compartment. In the nucleus, lncRNAs play several roles, including chromatin structure remodeling, epigenetic modulation, and regulation of transcription with lncRNA as enhancers or decoys [44]. We also found that DNA sequences in the CASC10 region bind transcription factors such as MYC, MAX, and CTCF (CCCTC-binding factor). Further studies using RNA fluorescence in situ hybridization (FISH), RNA tracking, and microscopic techniques are needed to fully understand the cellular localization and biological role of CASC10 in health and diseases. Likewise, we observed that several genes were regulated following CASC10 knockdown. In particular, high expression of Nucleoporin 43 (NUP43), one of the top downregulated genes in our study, is associated with DNA amplification and poor overall survival in luminal A and HER2+ breast cancer tumors [45]. Our observation that CASC10 knockdown arrested the cell cycle at the G0/G1 to S phase is in agreement with those reports as NUP43 plays a central role as a regulator of the mitotic progression and chromosome segregation [46]. One of the top increased transcript upon CASC10 knockdown was calponin-1 (CNN1) which encode a filament-associated protein with pivotal roles in cell metastasis, embryonic development, and prostate cancer progression [47]. Overexpression of CNN1 in breast cancer cells inhibited cell survival, migration, invasion, and enhanced apoptosis [48]. The observed increased expression of CNN1 could partially explain the observed increase in apoptosis following CASC10 knockdown.

Although this study focused on the biological, molecular, and therapeutic consequences of targeting CASC10, we identified several other clinically relevant transcripts in HGSOC. Future studies are needed to confirm their potential as targets for therapy and their contribution to the cisplatin resistance of HGSOC cells. Also, studies should be performed to clarify if a panel of these genes could be used to predict therapy response in HGSOC.

Overall, this study provides evidence that increased levels of the lncRNA CASC10 contribute to the cisplatin resistance of ovarian cancer cells and that a liposomal formulation of siRNA-CASC10 is a reasonable strategy for ovarian cancer treatment.

## 4. Materials and Methods

### 4.1. Cell Culture

The HGSOC cells OVCAR3 (NIH:OVCAR-3) and OV-90 were purchased from ATCC (Chicago, IL, USA). The human epithelial ovarian cancer cells SKOV3ip1 were a donation of Dr. Anil Sood (MD Anderson Cancer Center, Houston, TX). The cisplatin-resistant cells OVCAR3CIS, OV-90CIS, and SKOV3ip1CIS were generated by exposing cisplatin-sensitive cells to increasing doses of cisplatin. Briefly, cells were exposed to an initial concentration of 0.5 µM cisplatin (final concentration). One week later, the media was replaced by regular growth media. Cells were grown for an additional week and then exposed to 1.0 µM cisplatin (final concentration). This procedure was repeated, increasing the cisplatin concentration by 0.5 μM until a final concentration of around 10 µM. OVCAR3 and OVCAR3CIS were maintained in RPMI-1640 (HyClone, GE Healthcare Life Sciences, Logan, UT, USA) supplemented with 0.01 mg/mL insulin (Sigma-Aldrich, St. Louis, MO, USA), SKOV3ip1, and SKOV3ip1CIS cells were maintained in RPMI-1640 (HyClone), and OV-90 and OV90CIS were maintained on a 1:1 mixture of MCDB 105, and Medium 199 (Sigma-Aldrich). The culture media was supplemented with 10% Fetal Bovine Serum and 1% antibiotics at 37 °C in 5% CO_2_ and 95% air. All experiments were performed at 70–80% cell confluence.

### 4.2. RNA-Seq and Data Analysis of HGSOC Cells

Total RNA was isolated from OVCAR3, OVCAR3CIS, OV-90, and OV90CIS cells using the mirVana^TM^ miRNA Isolation Kit (Thermo Fisher Scientific, Grand Island, NY, USA) per the manufacturer’s instructions. RNA concentration and quality were verified on all samples with a Thermo Scientific NanoDrop spectrophotometer. RNA was enriched, and the library was prepared using GENEWIZ^®^ Strand-specific RNA sequencing with rRNA depletion (GENEWIZ, Inc. South Plainfield, NJ, USA). The library was quantified with KAPA SYBR^®^ FAST qPCR and then sequenced using the Illumina HiSeq (PE 2 × 150 bp) with a sequencing depth of approximately 100 million reads per sample. Unique gene counts were calculated using featureCounts from the Subread package (version 1.5.2, Parkville, Victoria, Australia), and initial gene expression analysis was performed using the DESeq2 (version 1.28.1) package in the R version 4.0.1 package.

### 4.3. Western Blot Analysis

Cell pellets were lysed with a complete lysis buffer (1% Triton X, 150 mmol/L NaCl, 25 mmol/L Tris HCl, 0.4 mmol/L NaVO_4_, 0.4 mmol/L NaF, and protease inhibitor cocktail from Sigma). Total protein concentration was quantified using the Bio-Rad DC Protein Assay reagents (Bio-Rad, Hercules, CA, USA). Protein samples were separated by SDS-PAGE, blotted onto nitrocellulose membranes, blocked in either 5% non-fat dry milk (Bio-Rad) or 5% BSA (HyClone), and probed with the appropriate dilution of the corresponding primary antibody. Membranes were then rinsed and incubated with the corresponding HRP-conjugated secondary antibody, followed by enhanced chemiluminescence and autoradiography. Bands were imaged with a Bio-Rad Gel Doc XR+, and signal intensity was quantified using Image Lab software (Bio-Rad, Hercules, CA, USA). The antibodies used are described in Appendix A. Western blot images with molecular weight markers are shown in Appendix A.

### 4.4. Small-Interfering RNA (siRNA) and In-Vitro Transfection

A siRNA ON-TARGET plus SMARTpool (a mixture of 4 siRNA as a single tube) was purchased from Horizon Discovery (Cambridge, UK). A negative control siRNA (NC-siRNA) (Sigma-Aldrich) was also used. To specifically target CASC10, two siRNAs targeting different regions of the CASC10 RNA (named CASC10-siRNA(1): 5′ GCUAUCUGCUUGAUCCCUU(dT)(dT) 3′ and CASC10-siRNA(2): 5′ GACUCUUGGAUCCAAGUUU(dT)(dT) 3′) were purchased from Sigma. OVCAR3CIS or SKOV3ip1CIS cells were seeded into 12-well plates at 3.0 × 10^4^ cells/mL. The next day, siRNAs were mixed with HiPerfect transfection reagent (Qiagen, Valencia, CA) at a 1:2 ratio (siRNA: HiPerfect) in serum and antibiotic-free Opti-MEM medium (Gibco, Thermo Fisher Scientific, Grand Island, NY, USA) and added to the cells. Twenty-four hours later, the media was replaced by regular culture media, and cells were cultured and used for further experiments. To assess siRNA efficiency, cells were collected 24 h after siRNA transfection.

### 4.5. Cell Growth and Cell Viability

To evaluate cell growth, we performed colony formation assays using Crystal Violet dye (Sigma-Aldrich). OVCAR3CIS cells (3.0 × 10^4^ cells/mL) or SKOV3ip1CIS (3.5 × 10^4^ cells/mL) were seeded into 12-well plates. Cells were transfected with 100 nM (final concentration) of each siRNA after twenty-four hours. The next day, 400 cells (OVCAR3CIS) or 1000 cells (SKOV3ip1CIS) were seeded into 6-well plates and incubated for seven days. Colonies were then fixed and stained with 0.5% crystal violet in methanol, and colonies of at least 50 cells were counted using a light microscope (CKX41; Olympus, Center Valley, PA, USA) with a total magnification of 40×. For cell viability assays, OVCAR3 and OVCAR3CIS (3.0 × 10^4^ cells/mL or SKOV3ip1 and SKOV3ip1CIS (3.5 × 10^4^ cells/mL) were seeded into 96-well plates, and 24 after hours, siRNA transfection was performed as described above. The following day, the transfection mix was replaced with cisplatin (CIS) (2.5 μM final concentration dissolved in regular cell culture media). Forty-eight hours after cisplatin treatment, the medium was removed, and Alamar blue dye (Invitrogen, Thermo Fisher Scientific, Eugene, OR, USA) was added following the manufacturer’s instructions. Optical density (OD) was measured using a plate reader (Bio-Rad), and percentages of cell viability were calculated after blank OD subtraction, taking the untreated cells as 100% cell viability.

### 4.6. Cell Invasion

Cell invasion was measured using the Matrigel transwell method as previously described [49,50]. OVCAR3CIS (3.0 × 10^4^ cells/mL) or SKOVip1CIS (3.5 × 10^4^ cells/mL) were seeded into 10 cm Petri dishes and transfected with siRNAs. After twenty-four hours, serum-free matrigel (BD Biosciences, San Jose, CA, USA) was added onto the upper chambers of 24-well plates and incubated at 37 °C for polymerization. Transfected cells were collected and resuspended in serum-free and re-seeded onto the Matrigel-coated chambers. Medium containing 10% FBS was added to the lower area of the wells, and the plates were incubated for 48 h at 37 °C. Then, the medium was removed, and cells that invaded through the matrigel were fixed and stained using the Protocol Hema 3 Stain Set (Fisher Scientific, Kalamazoo, MI, USA). Invaded cells were counted using the Olympus IX71 microscope equipped with a digital camera at a 20X resolution. The percentage of cell invasion was calculated using the NC-siRNA condition as 100% cell invasion.

### 4.7. Caspase-3 Activity

Caspase-3 activity was quantified using the Caspase-3/CPP32 Fluorometric Assay Kit (BioVision, Milpitas, CA, USA) as per the manufacturer’s instructions. OVCAR3CIS (3.0 × 10^4^ cells/mL) or SKOV3ip1CIS (3.5 × 10^4^ cells/mL) were seeded into 10 cm Petri dishes and transfected with NC-siRNA or CASC10-siRNA(2). After 24 h, the media was replaced by regular media, and seventy-two hours after transfection, cells were collected, pellets were lysed, and total protein concentration was determined. Equal amounts of protein were mixed with 2X Reaction Buffer and 1mM DEVD-AFC substrate in a 96-well plate and incubated at 37 °C for 2.5 h. Fluorescence intensity at 400 nm excitation and 505 nm emission was measured using the Varioskan Flash reader from Thermo.

### 4.8. Kaplan-Meier Survival Analysis

Kaplan-Meier (KM) patient survival analysis was performed using available patient datasets from gene chip and RNA-seq in the internet searchable database, Kaplan-Meier plotter (www.kmplot.com, accessed on 20 August 2018) [51]. For each gene, ovarian cancer patients were divided into high and low expression groups by the median value of their RNA expression. A set of different filters were applied in our search, including ovarian cancer patients, ovarian cancer patients treated with platinum, and serous ovarian cancer patients treated with platinum. Kaplan-Meier survival plots for overall survival (OS) and progression-free survival (PFS) were obtained with their respective hazard ratios (HR), confidence intervals (CI), and *p*-values (log-rank). For these studies, *p*-values < 0.05 were considered statistically significant.

### 4.9. KEGG Pathway Enrichment, Gene Ontology, and Network Analysis

Differentially abundant RNA transcripts with a log2 FC cutoff >1.2 or < −1.2 and a *p*-value ≤ 0.01 were selected for further assessment of their involvement in various biological pathways using the KEGG (Kyoto Encyclopedia of Genes and Genomes) pathway enrichment. Gene ontology and biological processes, molecular functions, and cellular components were enriched using Metascape.

For network analysis, we used Ingenuity Pathway Analysis (IPA; Ingenuity Systems, Qiagen, Redwood City, CA, USA) software to determine functional networks and pathways associated with differently abundant RNA transcripts using a *p*-value cutoff < 0.05.

### 4.10. SYBR-Green Based qRT-PCR

A custom-made 384-well plate containing pre-designed forward and reverse primers were purchased from Bio-Rad (Hercules, CA, USA). Total RNA was isolated from OVCAR3 and OVCAR3CIS cells using the GenElute Mammalian Total RNA Mini Kit (Millipore-Sigma, St. Louis, MO, USA) following the manufacturer’s instructions. RNA was reverse transcribed using the iScript Reverse Transcription Supermix for RT-qPCR from Bio-Rad. SYBR Green-based qPCR was performed using the SsoAdvanced™ Universal SYBR^®^ Green Supermix (Bio-Rad) and a CFX384 Touch Real-Time PCR detection system. Fold-changes and cycle threshold (Ct) values were calculated by the instrument’s internal software relative to OVCAR3 cells and normalized to β-actin along with controls for gDNA, PCR reaction, RT reaction, and RNA quality.4.11. Flow Cytometry.

To assess cell cycle progression, OVCAR3CIS (3.0 × 10^4^ cells/mL) or SKOV3ip1CIS (3.5 × 10^4^ cells/mL) were seeded into 10 cm Petri dishes and transfected with NC-siRNA or CASC10-siRNA(2), as described above. Forty-eight hours later, attached cells were collected, washed in ice-cold PBS, fixed with 70% cold ethanol, and stored at 4 °C. Cells were then washed with ice-cold PBS, resuspended in propidium iodide (PI)/RNase Staining Buffer (BD Biosciences, San Jose, CA, USA), incubated in the dark for 15 min at room temperature, and then analyzed by flow cytometry in BD C6 Accuri (San Jose, CA, USA). Accuri’s software was used to determine the percentage of cells in each cell cycle phase.

### 4.11. Tumor Implantation and Drug Treatment

Female athymic nude mice (NCr-nu, 6 weeks old) were purchased from Taconic (Hudson, NY, USA). To assess the therapeutic efficacy of liposomal CASC10-siRNA (CASC10-siRNA(2)) alone or in combination with Cisplatin (CIS) in vivo, mice were intraperitoneally (i.p.) injected with OVCAR3CIS (1.5 × 10^6^ cells/0.2 mL HBSS). After 7 days, mice were randomly divided into the following treatment groups (N = 10 per group): (a) NC-siRNA, (b) CIS alone, (c) CASC10-siRNA, (d) NC-siRNA plus CIS, and (e) CASC10-siRNA(2) plus CIS. Liposomal siRNAs (10 μg siRNA/injection) and CIS (160 μg/injection) were injected (i.p.) twice a week for four weeks. At the end of the treatment, mice were euthanized, tumors were collected, and the number of tumor nodules and tumor weight were recorded. Animal handling and research protocols were approved by the Institutional Animal Care and Use Committee (IACUC) of the University of Puerto Rico, Medical Sciences Campus.

### 4.12. RNA-Seq Analysis of CASC10-siRNA Transfected Cells

To prepare the RNA sequencing library, total RNA was extracted using the GenElute Mammalian Total RNA Miniprep Kit (Sigma). The RNA sample integrity was evaluated using a Bioanalyzer 2100 (Agilent Technologies, Santa Clara, CA, USA) (RNAs with a RIN > 7 were used). One μg of RNA was used for polyA mRNA enrichment. Library Prep was carried out on Poly A-selected samples using the NEXTFLEX-Rapid-Directional-RNA-Seq-Kit (Perkin Elmer, Waltham, MA, USA). The cDNA was then amplified for 15 PCR cycles (25 °C for 10 min, 42 °C for 15 min, 70 °C for 15 min, hold 4 °C). The PCR products were run on a NovaSeq 6000 S1 (Illumina, San Diego, CA, USA) flowcell running a 50-base paired-end (2X50) recipe. Preprocessing steps of quality, trimming, and filtering was performed using FastQCv0.11.9 and Trimmomaticv0.36 packages, applying the following parameters ILLUMINACLIP:2:15:10 LEADING:30 TRAILING:30 SLIDING-WINDOW:4:15 MINLEN:30. The trimmed and contamination-filtered reads were mapped to the hg38 genome using STAR aligner version 2.5.2a to generate a count matrix of the number of reads per identified gene. The differential expression analysis was carried out using the DESeq2 (version 1.28.1) package in the R version 4.0.1 package. As the count data was obtained in two batches, a batch correction term was introduced in the DESeq2 model using ComBat Seq to have better statistical power and control of false positives. Genes with a *p*-value < 0.01 after FDR adjustment and a log_2_ fold change >1 were considered significant for further analysis.

### 4.13. Statistical Analysis

All experiments were performed at least in triplicates. Graphing and statistical analysis were performed using the GraphPad Prism (San Diego, CA, USA) software 9.3.1. Data were analyzed using Student’s *t*-test for comparing two groups and ANOVA tests for multiple comparisons, with a significance level of 0.05.

## 5. Patents

A provisional Patent Application (No. 63/343,974) with data included in this manuscript was submitted to the United States Patent and Trademark Office (USPTO).

## Figures and Tables

**Figure 1 ijms-23-07737-f001:**
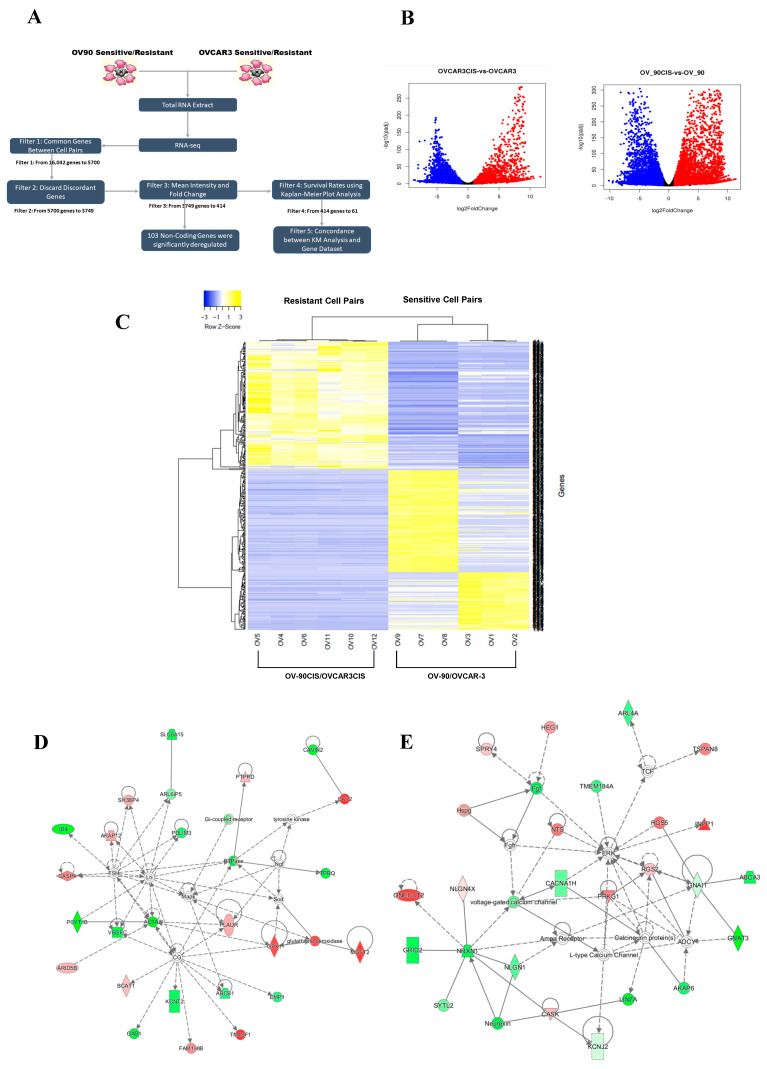
Identification of differentially abundant transcripts in cisplatin-sensitive and cisplatin-resistant HGSOC cells. (**A**) RNA-seq data filtering flowchart with the number of genes filtered in each step. (**B**) Volcano plots show the global transcriptional changes in the pairs of cell lines. The volcano plot (The log2FoldChange of each gene is represented on the x-axis, and the log10 of its adjusted *p*-value is on the y-axis. Genes with an adjusted *p*-value less than 0.05 and a log2FoldChange greater than one are indicated by red dots. These represent upregulated genes. Genes with an adjusted *p*-value less than 0.05 and a log2 fold change less than −1 are indicated by blue dots. These represent downregulated genes. (**C**) Heat map constructed with the 414 differentially abundant transcripts from filter-3. (**D**,**E**) Ingenuity pathway analysis (IPA). The top network (**left**) included molecules involved in cell survival pathways. The top second network (**right**) includes genes associated with vesicle trafficking, phosphorylation, and cGMP signaling. The green and red symbols denote downregulated and upregulated genes in the RNA-seq, respectively. Solid lines represent direct interactions between molecules, and dashed lines represent indirect interactions.

**Figure 2 ijms-23-07737-f002:**
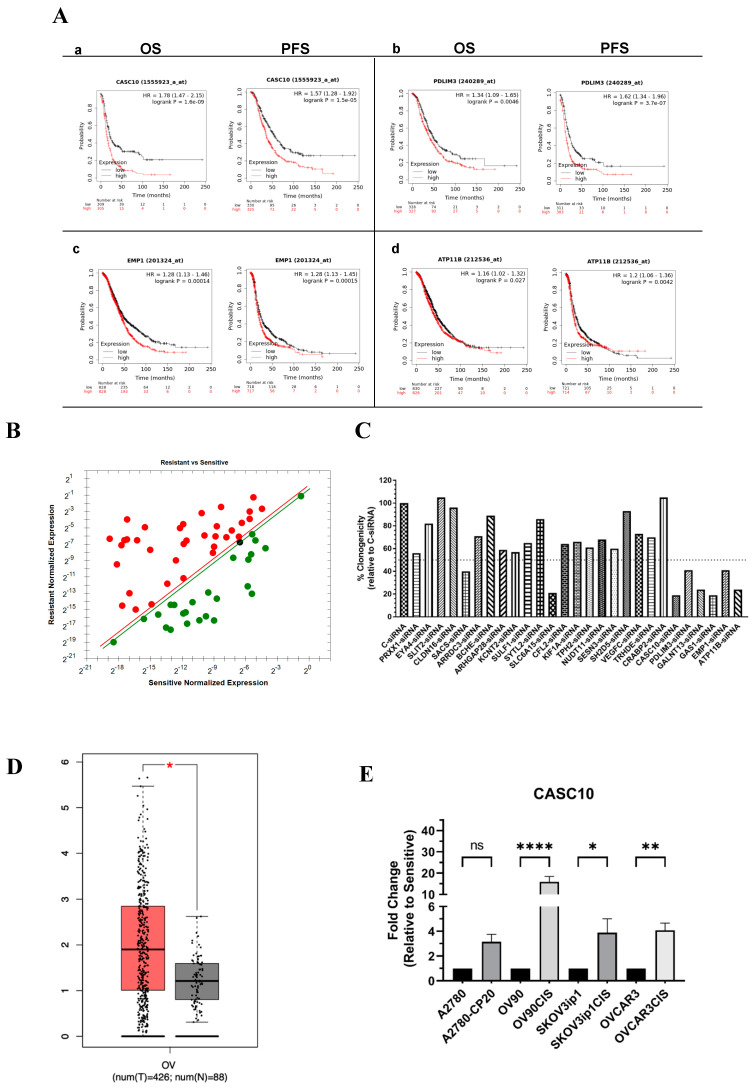
CASC10 is upregulated in ovarian cancer patients and cisplatin−resistant ovarian cancer cells. (**A**) Kaplan−Meier (KM) plots for gene expression−based overall survival (OS) and progression-free survival (PFS) analysis. KM plots of ovarian cancer patients were generated using the KM plotter searchable database. The OS and PFS of patients with ovarian cancer stratified by expression levels of (a) CASC10, (b) PDLIM3, (c) EMP1, and (d) ATP11B are shown based on gene chip analysis. *p*−values < 0.05 were considered to be statistically significant. (**B**) Validation of 45 differentially abundant transcripts by RT−qPCR in OVCAR3CIS cells. The normalized expression values were calculated relative to OVCAR3 (cisplatin−sensitive). Green and red symbols represent downregulated and upregulated genes, respectively. Diagonal green and red lines represent the selected threshold for significant fold changes. (**C**) siRNA screening for the 27 candidate genes. OVCAR3CIS cells were transiently transfected with siRNAs (100 nM), followed by a colony formation assay. The percent % of clonogenicity was expressed relative to the values obtained with the negative control siRNA (NC−siRNA). (**D**) Relative CASC10 expression levels in ovarian tumor tissues and normal ovarian tissues were analyzed using the GEPIA internet−searchable database. * *p* < 0.05. The red and black boxes represent cancer and normal tissue samples, respectively. (**E**) RT-qPCR of CASC10 relative expression in a panel of cisplatin−resistant versus cisplatin−sensitive ovarian cancer cells. (* *p* < 0.05, ** *p* < 0.01, and **** *p* < 0.0001).

**Figure 3 ijms-23-07737-f003:**
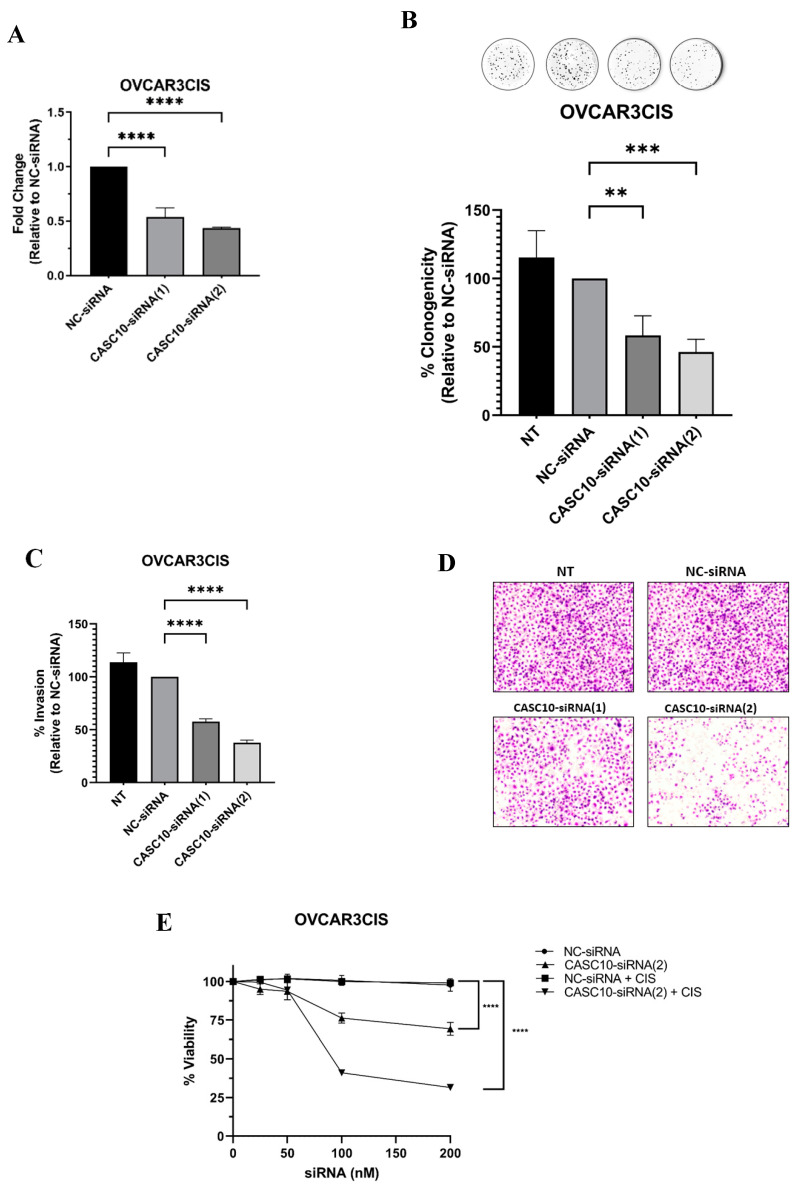
CASC10 siRNA-mediated knockdown reduced cell growth, invasion, and viability in ovarian cancer cells. (**A**) RT-qPCR following transfection of siRNAs in OVCAR3CIS cells. (**B**) colony formation assay, and (**C**,**D**) invasion ability following siRNA transfections in OVCAR3CIS cells. (**E**) Cell viability following siRNAs transfection in OVCAR3CIS cells. Cell viability was performed with and without CIS (2.5 μM). Experiments were performed at least in triplicates. Mean ± SEM is shown relative to NC-siRNA (** *p* < 0.01, *** *p* < 0.001, and **** *p* < 0.0001).

**Figure 4 ijms-23-07737-f004:**
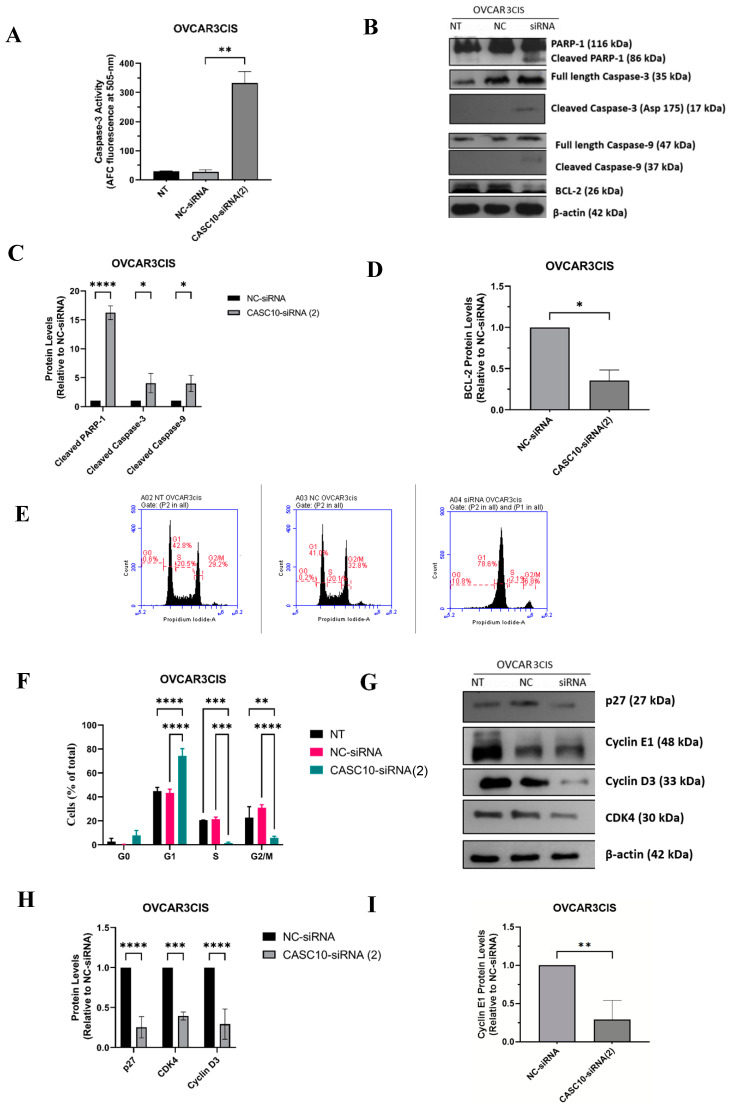
CASC10 siRNA-mediated knockdown Induced Apoptosis and Cell Cycle Arrest. OVCAR3CIS cells were transfected with 100 nM of NC-siRNA or 100 nM CASC10-siRNA(2). (**A**) Caspase-3 fluorometric activity assay in OVCAR3CIS cells 72 hr after transfection. (**B**) Western blot analysis of apoptotic-related proteins. (**C**,**D**) Densitometric analysis of the band intensities shown in (**B**). Mean ± SEM is shown relative to NC-siRNA (* *p* < 0.05, ** *p* < 0.01, *** *p* < 0.001, and **** *p* < 0.0001). (**E**) A histogram showing cell cycle arrest at G0/G1 to S phase transition after CASC10-siRNA(2) transfection in OVCAR3CIS cells compared to NC-siRNA. (**F**) Quantification of the flow cytometry data shown in (**E**). (**G**) Western blot analysis of cell cycle-related proteins 48 hr after siRNA transfection. (**H**,**I**) Densitometric analysis of the band intensities shown in (**G**). Mean ± SEM is shown relative to NC-siRNA (* *p* < 0.05, ** *p* < 0.01, *** *p* < 0.001, and **** *p* < 0.0001).

**Figure 5 ijms-23-07737-f005:**
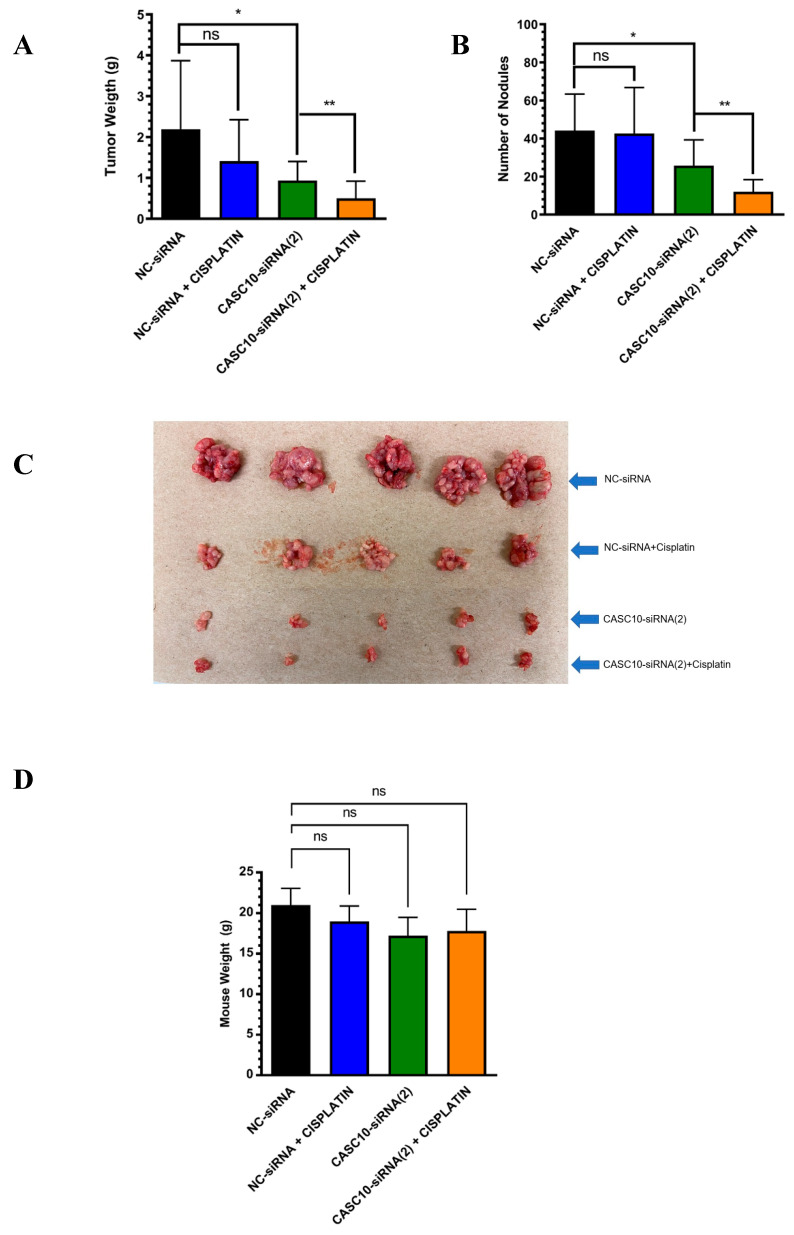
In vivo Targeting of CASC10 with liposome-encapsulated siRNAs. Therapeutic efficacy of liposomal CASC10-siRNA (CASC10-siRNA(2)) alone or in combination with Cisplatin (CIS) in vivo. Mice were intraperitoneally (i.p.) injected with OVCAR3CIS (1.5 × 10^6^ cells/0.2 mL HBSS). Seven days later, mice were randomly divided into the following treatment groups (N = 10 per group). Liposomal siRNAs (10 μg siRNA/injection) and CIS (80 μg/injection) were injected (i.p) twice a week for four weeks. Mean ± SEM is shown relative to NC-siRNA (* *p* < 0.05, ** *p* < 0.01 and ns = non significant). (**A**) tumor weight, (**B**) number of nodules, and (**C**) visual image of tumor size in all conditions. (**D**) Mice weight was recorded at the end of therapy.

**Figure 6 ijms-23-07737-f006:**
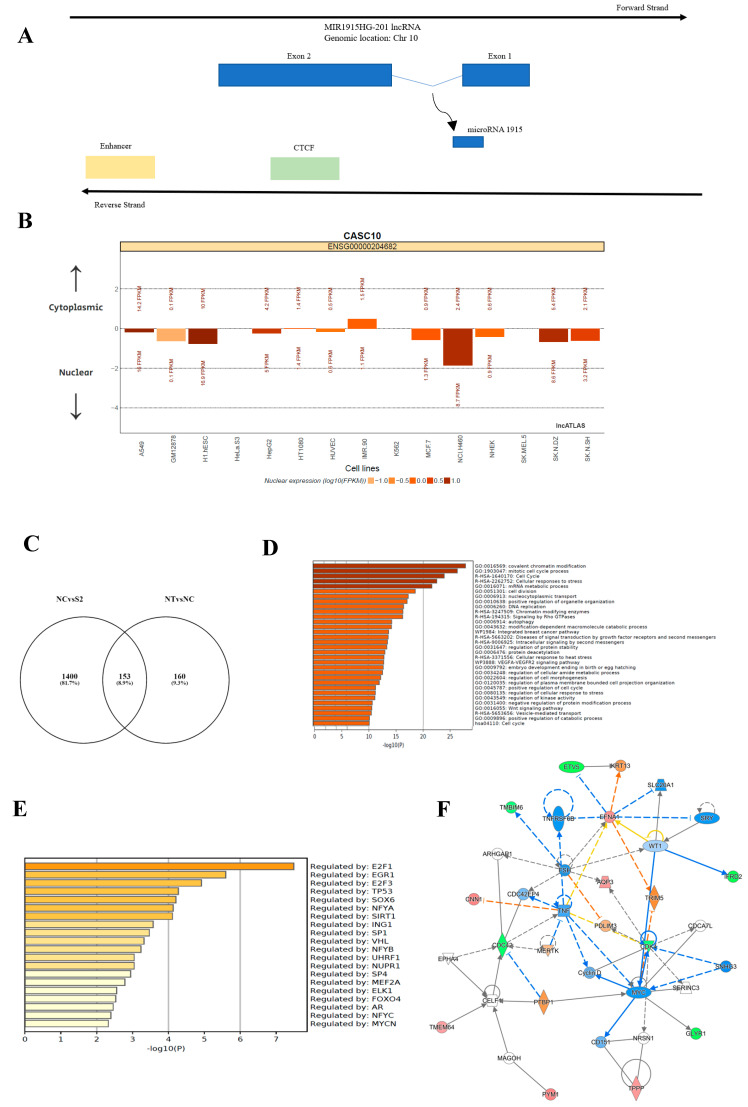
Downstream effectors of CASC10 in HGSOC cells. Transcriptome−wise analysis by RNA sequencing (RNA−seq) following siRNA−mediated CASC10 knockdown in OVCAR3CIS cells. (**A**) Genomic information of CASC10 (flanking sequences), chromosomal location, and RNA structure using the NCBI and ENSEMBL databases. (**B**) CASC10 subcellular localization plots displayed by lncATLAS. Bars representing CN-RCI values for CASC10 across all cell lines. Expression values (FPKMs) for CASC10 are shown for both compartments (cytoplasm on top of the bar and nucleus on the bottom). Bars are colored by their absolute nuclear expression. (**C**) Venn diagram showing that 1400 differentially abundant RNA transcripts in NC−siRNA vs. CASC10−siRNA(2) in OVCAR3CIS cells. (**D**,**E**) Gene ontology and KEGG analysis of functional enrichment via Metascape. The 20 top most significant (*p*-value ≤ 0.01) enriched ontology clusters (**D**) and the top 20 most enriched ontology clusters regulated by transcription factors (**E**). (**F**) Ingenuity Pathway Analysis (IPA) following siRNA−mediated CASC10 knockdown. The top network is shown and involved in cell cycle regulation, programmed cell death, and survival-related genes.

**Table 1 ijms-23-07737-t001:** Relative expression values of the 45 differentially abundant RNA transcripts in OVCAR3CIS vs. OVCAR3 cells.

Gene	qPCR Log2FC	RNA-Seq Log2FC
*PDLIM3*	13.16	5.56
*TPH2*	11.53	8.04
*TRHDE*	10.67	7.82
*KCNT2*	10.47	5.09
*GAS1*	10.24	5.01
*CLDN16*	9.33	5.29
*PRRX1*	9.09	4.31
*SESN3*	9.07	3.41
*GALNT13*	8.50	4.78
*SYTL2*	7.63	4.55
*SULF1*	7.43	3.78
*BCHE*	6.50	1.31
*ATP11B*	6.44	3.24
*SLIT2*	5.33	2.09
*SLC6A15*	5.29	5.45
*SH2D5*	5.19	2.51
*VEGFC*	4.85	4.84
*CFL2*	4.42	1.98
*ARRDC3*	4.17	2.24
*CRABP2*	3.43	1.62
*NUDT11*	3.34	2.13
*EYA4*	2.74	0.58
*SACS*	2.55	1.03
*KIF1A*	2.36	1.43
*ARHGAP28*	2.19	1.24
*EMP1*	2.11	1.20
*DDAH1*	2.09	0.96
*CASC10*	2.06	0.50
*LRRC17*	−0.94	−1.42
*LRG1*	−0.97	−2.72
*TMCC3*	−1.03	−2.25
*SPRY4*	−1.67	−1.91
*F2R*	−2.09	−3.80
*SULT1A1*	−2.67	−3.43
*PBX1*	−2.78	−2.29
*SLC7A2*	−2.85	−5.47
*MXRA8*	−3.16	−4.14
*NID1*	−3.36	−4.50
*RUNX1*	−3.59	−1.64
*PROCR*	−4.28	−5.77
*LIPG*	−5.20	−5.68
*PDE1A*	−5.76	−7.53
*PTPRD*	−6.38	−7.88
*VCAN*	−6.60	−5.64
*THBS1*	−9.79	−4.06

**Table 2 ijms-23-07737-t002:** Top five upregulated and top five downregulated genes in CASC10-siRNA(2) vs. NC-siRNA.

Gene Symbol	Gene Name	Biological Role	*p*-Value
Upregulated			
RTN4R	Reticulon 4 receptor	Receptor-mediated axonal growth inhibition	4.59 × 10^−21^
MACF1	Microtubule actin crosslinking factor 1	Actin-microtubule interactions	2.83 × 10^−12^
PYM1	PYM homolog 1	Nuclear-transcribed mRNA catabolic process and regulation of translation	5.24 × 10^−20^
CNN1	Cellular communication network factor 1	Regulation of vascular-associated smooth muscle cell proliferation	7.38 × 10^−6^
TGFBRAP1	Transforming growth factor beta receptor associated protein 1	TGF-beta signaling and association to SMAD4	3.32 × 10^−39^
Downregulated			
NUP43	Nucleoporin 43	Transport of macromolecules between cytoplasm and nucleus	1.69 × 10^−86^
FHL1	Four and a half LIM domains 1	Assembly of sarcomeres and muscle contraction regulation	2.49 × 10^−11^
DHFR2	Dihydrofolate reductase 2	Tetrahydrofolate metabolic process and thymidine biosynthesis	1.21 × 10^−10^
MIR1915HG	microRNA1915 host gene	Long Noncoding RNA	2.76 × 10^−59^
NDUFA7	NADH:ubiquinone oxidoreductase subunit A7	Member of complex I in the electron transport chain	7.76 × 10^−29^

## Data Availability

The data presented in this study is available on request from the corresponding author.

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
