# Peer review of "Upregulation of the Long Noncoding RNA CASC10 Promotes Cisplatin Resistance in High-Grade Serous Ovarian Cancer"

_ijms, 2022, doi:10.3390/ijms23147737_

Round 1

Reviewer 1 Report

This is a very interesting study supporting the role of CASC10 in the development of platinum resistance in ovarian carcinomas. Authors provide adequate experimental data. 

However, I believe that mouse model data are not in line with their conclusions. CASC10siRNA indeed decreased mobility, invasiveness and viability of ovarian cancer platinum resistant cell lines. However, no synergistic effect with platinum treatment is obvious and this conclusion should be removed from the conclusion.

TO document that CASC10 overexpression is a mechanism of cisplatin resistance with clinical consequences, I believe that authors should prove that CASC10silencing in platinum sensitive cells does not affect viability and invasion. 

Author Response

Reviewer 1

This is a very interesting study supporting the role of CASC10 in the development of platinum resistance in ovarian carcinomas. Authors provide adequate experimental data. However,

  1. I believe that mouse model data are not in line with their conclusions. CASC10siRNA indeed decreased mobility, invasiveness and viability of ovarian cancer platinum resistant cell lines. However, no synergistic effect with platinum treatment is obvious and this conclusion should be removed from the conclusion.

R/. In the Discussion section of the revised manuscript, we better explain our findings regarding the therapeutic effects of combining CASC10-siRNA with cisplatin. Lines 415-418. In fact, we included a Figure that was not included in the original submission regarding the number of nodules of the therapeutic experiment (Figure 4B, revised version). We observed a reduction in the number of nodules in the CASC10-siRNA group compared with the NC-siRNA. Also, cisplatin exacerbated the CASC10-siRNA effect.

  1. To document that CASC10 overexpression is a mechanism of cisplatin resistance with clinical consequences, I believe that authors should prove that CASC10 silencing in platinum sensitive cells does not affect viability and invasion.

R/. We performed these experiments in OVCAR3 and SKOVip1 cells. The results were included as part of the Figure S2 F-I in the revised version of the manuscript.

Reviewer 2 Report

In this study authors found that the lncRNA CASC10  is  upregulated in cisplatin-resistant ovarian cancer cells and ovarian cancer patients. Moreover, CASC10 knockdown significantly reduced cell proliferation and invasion; and sensitized cells to cisplatin treatment inducing apoptosis and impairing the expression of several CASC10 downstream effectors.

The manuscript is clear and generally well written but it presents some points that need to be improved before publication. In particular: 

Line 44: authors should describe more deeply the mechanism of action of cisplatin; indeed, they state that "Cisplatin forms adducts with the DNA, which causes the inhibition of replication and transcription, and leads to cell-cycle arrest and apoptosis". This statement is excessively simplified. Cisplatin becomes active upon entering the cell and once in the nucleus, it binds to the purine N7 atom triggering DNA damage in cancer cells, blocking cell division, and resulting in apoptotic cell death. Another key mechanism involved in cisplatin toxicity is the oxidative stress. Indeed, mitochondria are the major target for the oxidative stress that is cisplatin-induced, resulting in the loss of the mitochondrial protein sulfhydryl group, preventing calcium uptake and triggering a decrease of the mitochondrial membrane potential (PMID: 32093309; PMID: 35453297). All these considerations should be reported by authors since their findings are crucially linked to the cisplatin mechanisms of action.

Figure 1,2: improve image quality since it is not readable. 

Figure 3: improve image quality 

Line 116: in table 1 are shown 16 (not 17) downregulated genes. Please correct

Line 354: Reference 26 is not appropriated since it is referred to bladder cancer. It should be replaced with a more specific literature since at least 3 important pathways plays a key role in cisplatin chemoresistance in ovarian cancer as also recently reviewed (PMID: 35582310, 35453348, 35383278).

Line 449: a more detailed method of cisplatin resistance induction must be reported

References and citations: Authors must follow the journal style

An accurate revision of typing errors is recommended

Author Response

The manuscript is clear and generally well written but it presents some points that need to be improved before publication. In particular:

  1. Line 44: authors should describe more deeply the mechanism of action of cisplatin; indeed, they state that "Cisplatin forms adducts with the DNA, which causes the inhibition of replication and transcription, and leads to cell-cycle arrest and apoptosis". This statement is excessively simplified. Cisplatin becomes active upon entering the cell and once in the nucleus, it binds to the purine N7 atom triggering DNA damage in cancer cells, blocking cell division, and resulting in apoptotic cell death. Another key mechanism involved in cisplatin toxicity is the oxidative stress. Indeed, mitochondria are the major target for the oxidative stress that is cisplatin-induced, resulting in the loss of the mitochondrial protein sulfhydryl group, preventing calcium uptake and triggering a decrease of the mitochondrial membrane potential (PMID: 32093309; PMID: 35453297). All these considerations should be reported by authors since their findings are crucially linked to the cisplatin mechanisms of action.

R/.       In the Introduction section of the revised manuscript, we included additional information regarding the mechanisms of action of cisplatin. Lines 43-52.

  1. Figure 1,2: improve image quality since it is not readable.

R/.       We improved the image size and quality of Figures 1 and 2 in the new version of the manuscript.

  1. Figure 3: improve image quality

R/.       We improved the image size and quality of Figure 3 in the new version of the manuscript.

  1. Line 116: in table 1 are shown 16 (not 17) downregulated genes. Please correct

R/.       In the new version of the manuscript, we included an updated, revised version of Table 1 with the corresponding number of downregulated genes.

  1. Line 354: Reference 26 is not appropriated since it is referred to bladder cancer. It should be replaced with a more specific literature since at least 3 important pathways plays a key role in cisplatin chemoresistance in ovarian cancer as also recently reviewed (PMID: 35582310, 35453348, 35383278).

R/.       We included an appropriate reference for this section in the new version of the manuscript. Line: 372.

  1. Line 449: a more detailed method of cisplatin resistance induction must be reported

R/.       In the Methods section of the revised manuscript, we included a more detailed explanation of how cisplatin-resistant cells were generated. Lines 457-463.

  1. References and citations: Authors must follow the journal style

R/.       All references were corrected following the journal style in the new version of the manuscript.

  1. An accurate revision of typing errors is recommended

R/.       The manuscript was submitted for editing, and typing errors were corrected in the new version of the manuscript.

Round 2

Reviewer 2 Report

The manuscript has been significantly improved and can be accepted in the present form